# A Systematic Review and Meta-analysis of the Prevalence and Impact of Pulmonary Bacterial Colonisation in Stable State Chronic Obstructive Pulmonary Disease (COPD)

**DOI:** 10.3390/biomedicines10010081

**Published:** 2021-12-31

**Authors:** Michael N. Armitage, Daniella A. Spittle, Alice M. Turner

**Affiliations:** 1Medical Education, University Hospitals Coventry and Warwickshire NHS Trust, Clifford Bridge Rd., Coventry CV2 2DX, UK; Michael.armitage@doctors.org.uk; 2Institute of Inflammation and Ageing, University of Birmingham, Birmingham B15 2TT, UK; d.a.spittle@bham.ac.uk; 3University Hospitals Birmingham NHS Foundation Trust, Institute of Applied Health Research, University of Birmingham, Birmingham B15 2TT, UK

**Keywords:** COPD, chronic bronchitis, emphysema, stable, colonisation, prevalence

## Abstract

Background: Half of acute exacerbations of COPD are due to bacterial infection, and the other half are likely influenced by microbial colonisation. The same organisms commonly cultured during acute exacerbations are often found in the sputum of patients during stability. A robust assessment of the prevalence of potentially pathogenic microorganisms (PPMs) in the sputum of stable COPD patients may help to inform the targeted prevention of exacerbation by these organisms. Methods: A systematic review and meta-analysis was carried out to determine the prevalence of PPMs in patients with COPD in the stable state. Meta-analysis of prevalence was carried out using the Freeman–Tukey double arcsine transformation random effects model, and sub-group analysis was performed for sputum modality. Prevalence of total and individual PPMs was calculated from patient-level data from individual studies. Results: Pooled prevalence of PPMs identified by sputum culture was found to be 41% (95% CI 36–47%). Significant heterogeneity was found across all studies, which can likely be attributed to inconsistent measuring and reporting of PPMs. The most commonly reported organisms were *H. influenzae*, *M catarrhalis*, *S. pneumoniae*, *S. aureus*, and *P. aeruginosa*. Declining lung function was weakly correlated with prevalence of PPMs. Conclusion: The airways of patients with COPD are colonised with PPMs during the stable state in almost half of patients. A complex relationship likely exists between the microbiome in the stable state and the phenotype of COPD patients. Targeted microbial therapy for preventing exacerbations of COPD should carefully consider the stable microbiome as well as the exacerbated.

## 1. Introduction

Colonisation of the lung by potentially pathogenic microorganisms is a classical feature of chronic obstructive pulmonary disease (COPD) [1]. The residence of potentially pathogenic bacteria, as opposed to commensal, is described as bacterial colonisation and is associated with persistent, neutrophil-mediated inflammation. From a clinical perspective, bacterial colonisation has been shown to cause worsening of symptoms, an increased frequency of exacerbations, and to accelerate lung function decline. COPD patients with bacterial colonisation also show higher daily sputum production, dyspnoea, and cough [2].

Approximately 50% of acute exacerbations of COPD are caused by bacterial infections, with non-typeable *Haemophilus influenzae*, *Moraxella catarrhalis*, *Streptococcus pneumoniae*, and/or *Pseudomonas aeruginosa* being the most common isolates [3]. These microbes are also the most commonly reported as colonising the airways of COPD patients while in the stable state [4]. However, the local environment within the respiratory tract can be sampled in several ways; through spontaneous sputum, induced sputum, bronchoalveolar lavage (BAL), or protected specimen brushing (PSB) (the latter two being obtained via bronchoscopy). Each of these sample types will reflect different areas of the respiratory tract and will display varied physiology and microbiological communities [5,6].

The use of antibiotics for the prophylactic treatment of infective exacerbations is widely recognised. Prophylactic regimens typically include either continuous or intermittent treatment of macrolides (e.g., azithromycin), tetracyclines (e.g., doxycycline), beta-lactams (e.g., penicillin), or fluoroquinolones (e.g., moxifloxacin). While the choice of treatment varies globally, which renders it difficult to assess the efficacy of prophylaxis in real-life care in COPD, the use of prophylactic antibiotics in randomised controlled trials has been shown by meta-analysis to reduce the number of patients experiencing one or more exacerbation, as well as individual exacerbation frequency. [7] However, not all patients respond to antimicrobial prophylaxis and it has been recommended that future work consider the potential role that microbial colonisation may play in determining this response [7].

In this review and meta-analysis, we seek to investigate the prevalence of bacterial colonisation in stable COPD with the intent of informing future targeted therapies for the prevention of COPD exacerbation and deterioration. As secondary aims, we intend to illustrate the effects of sputum sample modality on bacterial prevalence and to assess the relationship between colonisation and disease phenotype.

## 2. Methods

This systematic review and meta-analysis was conducted in accordance with the Preferred Reporting Items for Systematic Reviews and Meta-Analysis (PRISMA) guidelines, and the protocol has been registered on the NIHR PROSPERO database with ID number CRD42021254950.

### 2.1. Search Strategy

Included databases were MEDLINE and EMBASE through Ovid, and CINAHL through EBSCO. CENTRAL and ClinicalTrials.gov were also searched for relevant ongoing clinical trials. Databases were searched for results from conception to June 2021. Searches were limited to human participants and were not limited by language. Primary and secondary outcomes are detailed in Table 1. The full search strategy is shown in the Appendix A. Only those studies that reported the primary outcome were included. Studies of patients reported as being in the stable state with a clinical diagnosis of chronic obstructive pulmonary disease (COPD), chronic bronchitis, or emphysema were included. Studies with mixed populations that have separately reported data on a subgroup of patients who fit these criteria were also included. Randomised controlled trials (RCTs) were included if they reported the primary outcome at randomisation or in a control arm. Studies were excluded if they enrolled patients who were currently exacerbated, were under the age of 18, or had a history of bronchiectasis or malignancy. Exclusion criteria included patients with a recorded history of bronchiectasis and COPD patients with a recent exacerbation (as defined by the primary work; see also discussion), and/or patients that had a recorded history of treatment for any cancer diagnosis.

### 2.2. Study Selection, Data Extraction and Quality Assessment

Studies were screened initially by title and abstract, and then through full text, against the inclusion and exclusion criteria by two independent reviewers (MNA and DAS). Disagreements were resolved through discussion between authors. Non-English language papers were screened by one author (AMT) with the assistance of a native language speaker. Data were extracted through the use of a piloted and standardised spreadsheet. To avoid misrepresentation, where data were not mentioned in the text, fields were marked as “n”. If the data point had been sought, but not found, “0” was entered into the field. Data were extracted for all bacteria reported as PPMs in the text including % positive samples and where available, total bacterial count. Where patient-level colonisation data were reported, summary statistics were performed for the study cohort. If studies contained 2 subgroups that both met the inclusion criteria, data were collected separately and studies were labelled as “Author, a” and “Author, b”. Authors of papers with incomplete or uncertain findings were contacted for further information. Papers were excluded if results were not obtainable or if an unsatisfactory response was received. At full-text screen, papers were excluded if the primary outcome was not reported. Risk of bias (quality) assessment was performed by two independent authors (MNA and DAS), using the Joanna Briggs Institute (JBI) checklist for prevalence studies critical appraisal tool [8].

### 2.3. Statistical Analysis

A meta-analysis of prevalence data was carried out on the prevalence of bacterial colonisation across all studies in accordance with the methods described by Barker et al., [9]. The Freeman–Tukey double arcsine transformation random effects model was applied. Subgroup analysis was performed following the same method with studies grouped by sputum collection modality. Individual study prevalence confidence intervals were calculated from patient-level data using the Clopper-Pearson method. All statistical analyses were performed in R statistical software (v4.1.1) using the ‘metafor’ package for meta-analysis. 

## 3. Results

Overall, 4568 studies were retrieved through the initial search strategy with 257 studies reaching full text screening. A total of 36 studies were included for data extraction and meta-analysis (Figure 1). The most common reason for exclusion at full-text screen was the absence of reporting our primary outcome (*n* = 96). Most included studies were prospective cohort or cross-sectional studies, with the majority of populations consisting of stable COPD outpatients. Several studies included subgroups of COPD patients (i.e., moderate vs. advanced disease), and where this was the case data have been collected separately for each subgroup. There was a variety of modalities used for sputum collection, with some studies employing multiple methods. No studies were retrieved which specified collection from patients with alpha-1 antitrypsin (A1AT) deficiency. A summary of study characteristics and population demographics can be found in Table 2. The JBI tool highlighted small sample sizes across studies as the main risk of bias (Appendix A). Several studies were also influenced by inconsistent reporting of PPMs, or low response rate (not all patients producing sputum samples).

The 36 studies included 3576 participants, of which 2809 had stable COPD and were included for analysis. Mean age across studies was 65.7 years, and studies averaged 22.7% female. The majority of studies required patients to be in stable condition for a minimum of 4 weeks, with criteria ranging from 15 days to 16 weeks. Some studies required only that patients be ‘in stable condition’ without quantifying this. Across studies 34.6% of patients were current smokers, 64.3% were ex-smokers, and 1.1% were never smokers. Across those studies that reported these outcomes, mean pack-years were 50.9, and mean yearly COPD exacerbations were 1.9 (although exacerbation data were poorly reported across studies).

For the purpose of meta-analysis, intra-study sub-groups were clustered. A large amount of heterogeneity was observed between the studies (I^2^ = 94%). Sample size varied across the 36 studies with the smallest study containing only 14 samples [10] and the largest, 1628 samples [11]. A total of 11,862 samples were retrieved from 2425 patients across the 36 studies (Figure 2). Of these samples, a total of 3900 grew a culture positive for a PPM (41%, 95% CI 36–47%) (Table 3, Figure 2). Only one study reported 100% prevalence, and all studies reported at least one positive culture. Data on bacterial load in positive cultures were sparingly and inconsistently reported. 

Sampling modality differed across the 36 studies; 7 collected spontaneous sputum samples, 6 induced sputum samples, 7 spontaneous and induced sputum samples, 7 protected-specimen brush samples, 10 bronchoalveolar lavage samples, and 1 trans-tracheal aspiration. There was a pooled mean prevalence of 44% (95% CI 36–53%; I^2^ = 92%) in the spontaneous sputum subgroup, 45% (95% CI 29–61%; I^2^ = 91%) in the induced sputum subgroup, 38% (95% CI 28–49%; I^2^ = 92%) in combined spontaneous and induced sputum subgroup, 27% (95% CI 22–32%; I^2^ = 0%) in the PSB subgroup, 48% (95% CI 34–62%; I^2^ = 85%) in the BAL subgroup and 33% (95% CI 11–59%; I^2^ = N/A) in the trans-tracheal aspiration subgroup (Figure 3). *H. influenzae* was the most commonly isolated organism for each sampling modality (Table 3). 

FEV1 data were reported across most studies as mean FEV1 (% predicted), with three studies reporting median, and two studies reporting no pulmonary function test results. The mean FEV1 (% predicted) across those studies reporting this data was 53.6, with the majority of patients reported as either GOLD criteria stage 2 (36.0%), or stage 3 (27.8%). There was a weak correlation between FEV1 and colonisation (R^2^ = 0.136, *p* = 0.015, Figure 4). Outcome data including quality of life, hospitalisation, and mortality were poorly reported across all studies, with no studies reporting mortality.

The only PPM reported in all 36 (100%) studies was *H. influenzae*, with the next most commonly reported PPMs being *M catarrhalis* (*n* = 34, 94%)*, S. pneumoniae* (*n* = 33, 92%), *S. aureus* (*n* = 24, 67%), and *P. aeruginosa* (*n* = 27, 75%) (Figure 5). There was a general trend across studies towards *H. influenzae* as the most prevalent organism.

**Table 2 biomedicines-10-00081-t002:** Study characteristics and population demographics.

Study (Author, Year)	Country	Study Design	Population Description	Study Subgroup	No. Stable COPD Patients (% Female)	Age	Smoking, Pack-Years	FEV1, % Predicted	Stability Period Pre-Sampling
Andelid et al., 2015 [12]	Sweden	Prospective cohort	Smokers with obstructive disease and chronic bronchitis	-	60 (24)	62 {45–76}	40 {14–156}	60 {29–97}	15 weeks
Banerjee et al., 2004 [13]	UK	Prospective cohort	Stable COPD outpatients	-	67 (NA)	66.7 (7.6)	58.8 (25.1)	43.2 (11.4)	6 weeks
Bogaert et al., 2004 [14]	Netherlands	Prospective cohort	Stable COPD outpatients	-	269 (NA)	{40–75}	NA	NA	Stable clinical condition
Cabello et al., 1997 [1]	Spain	Prospective cohort	Stable COPD outpatients with indication for bronchoscopy	PSB	18 (17)	60 (12)	NA	77 (19)	4 weeks
				BAL	18 (17)	60 (12)	NA	77 (19)	4 weeks
Einarsson et al., 2016 [15]	UK	Cross-sectional	COPD patients listed for bronchoscopy	-	18 (22)	60 {41–74}	NA	57 {32–89}	8 weeks
Fruchter et al., 2014 [16]	Israel	Prospective cohort	Severe COPD pre-BLVR	-	70 (22)	64 (8)	28 (11)	34.6 (7.3)	90 days
Garcia-Nunez et al., 2014 [17]	Spain	Cross-sectional	Stable COPD outpatients	Moderate-to-severe disease	17 (13)	68 {62–69}	75 {52–110}	52.0 {41.5–69.0}	4 weeks
				Advanced disease	17 (0)	74 {68–77}	55 {35–117}	32.0 {29.5–35.0}	4 weeks
Hurst et al., 2005 [18]	UK	Prospective cohort	Stable COPD outpatients	Whole cohort	47 (43)	70.5 (7)	46.1 (26.5)	37.9 (13.6)	12 weeks
Jacobs et al., 2018 [19]	USA	Prospective cohort	Stable COPD outpatients	-	181 (NA)	67 (9.2)	79 (36)	49 (18)	Stable clinical condition
Jordan et al., 1976 [20]	USA	Cross-sectional	Chronic bronchitis patients	BAL	19 (NA)	NA	NA	NA	Stable clinical condition
				Trans-tracheal aspiration	19 (NA)	NA	NA	NA	Stable clinical condition
Khurana et al., 2014 [21]	UK	Cross-sectional	Stable COPD outpatients	Non-persistent sputum	52 (46)	66.8 (6.5)	35.3 {12.5–86}	65.1 (16.3)	6 weeks
				Persistent sputum	52 (54)	65.7 (6.9)	32.0 {18.5–122.2}	54.5 (13.1)	6 weeks
Marin et al., 2009 [22]	Spain	Prospective cohort	Stable COPD outpatients	Baseline	40 (3)	66.5 (8.1)	NA	57.9 (19.1)	8 weeks
				9 month follow-up	40 (3)	66.5 (8.1)	NA	57.9 (19.1)	8 weeks
Marin et al., 2012 [23]	Spain	Cross-sectional	COPD recruited on hospitalization for exacerbation	-	133 (7)	70 (9)	67 {43–102}	52 (16)	12 weeks
Mika et al., 2018 [24]	Switzerland	Cross-sectional	COPD patients listed for bronchoscopy	-	32 (31)	65.7 (NA)	NA	50.2 (24.9)	Stable clinical condition
Millares et al., 2014 [10]	Spain	Prospective cohort	COPD patients with >2 exacerbations per year	Whole cohort	16 (0)	71 (6)	57 {57–110}	36 {30–40}	>8 weeks
Miravitlles et al., 2009 [25]	Spain	Randomised control trial	COPD with sputum positive for PPM (p. aeruginosa excluded)	At randomisation	119 (8)	68 (9.1)	NA	46.2 (14.1)	16 weeks
				Placebo 8 week follow-up	119 (5)	69 (10)	43 (21)	53 (16)	16 weeks
Miravitlles et al., 2010 [26]	Spain	Cross-sectional	Stable COPD outpatients	-	119 (6)	68 (9.1)	40 (21.1)	46.4 (14.1)	12 weeks
Monso et al., 1995 [27]	Spain	Cross-sectional	COPD patients listed for bronchoscopy	-	40 (0)	61.1 (9.9)	NA	51.2 (23)	15 days
Monso et al., 1999 [28]	Spain	Cross-sectional	Stable chronic bronchitis	-	41 (0)	63.8 (9.1)	NA	74.6 (23.7)	15 days
Patel et al., 2002 [4]	UK	Prospective cohort	Stable COPD outpatients	-	29 (28)	66 {47–81}	52.9 (42.2)	38.7 (15.2)	3 weeks
Riise et al., 1994 [29]	Sweden	Prospective cohort	Chronic bronchitis with and without COPD	Without COPD	41 (NA)	52 {36–68}	36, 2 *	92, 2 *	4 weeks
				With COPD	41 (NA)	57 {38–70}	44, 4 *	62, 2 *	4 weeks
Seemungal et al., 2008 [30]	UK	Randomised control trial	Stable COPD outpatients at baseline	-	109 (37)	67.2 (8.6)	51.6 (33.9)	50.0 (18.0)	4 weeks
Sethi et al., 2006 [31]	USA	Prospective cohort	Ex-smokers with COPD	-	26 (23)	64.7 (1.7)	66 (6.3)	59.8 (4.1)	4 weeks
Sibila et al., 2014 [32]	Spain	Cross-sectional	Stable COPD outpatients	-	37 (24)	67.9 (8.0)	47.3 (12.7)	40.9 (8.1)	4 weeks
Sibila et al., 2016 [33]	Spain	Cross-sectional	Stable COPD outpatients	-	45 (18)	67.1 (8.5)	54.3 (20.1)	41.3 (10.2)	4 weeks
Simpson et al., 2014 [34]	Australia	Randomised control trial	Stable COPD outpatients at randomisation	-	30 (37)	70.8 (7.6)	46.1 (36.6)	53.7 (13.7)	4 weeks
Simpson et al., 2016 [35]	Australia	Cross-sectional	Stable COPD outpatients	-	59 (51)	69.7 (7.5)	32.9 {17.0–53.8}	54.3 (15.6)	Stable clinical condition
Singh et al., 2014 [36]	UK	Prospective cohort	Stable COPD outpatients	-	99 (33)	72.1 (8.9)	48.4 {24.4–67.5}	51.5 (21.6)	4 weeks
Sriram et al., 2018 [37]	Australia	Cross-sectional	COPD patients listed for bronchoscopy	-	27 (37)	68 (9)	43 (28)	68 (25)	Excluded exacerbations
Trudzinski et al., 2018 [38]	Germany	Cross-sectional	COPD patients undergoing BLVR with EBV insertion	-	64 (50)	62.4 (8.7)	NA	27.3 (9.5)	Excluded exacerbations
Tumkaya et al., 2006 [39]	Turkey	Prospective cohort	Stable COPD outpatients	Exacerbations (<3/year)	39 (10)	58.6 (7.7)	46.2 (22.1)	70.5 (12.0)	4 weeks
				Exacerbations (>3/year)	39 (11)	58.8 (7.7)	50.26 (22.2)	65.8 (12.8)	4 weeks
Weinreich et al., 2007 [40]	Denmark	Cross-sectional	COPD patients listed for bronchoscopy	-	53 (49)	67 {58–73}	30 {21–45}	44 {NA}	4 weeks
Wilkinson et al., 2003 [41]	UK	Prospective cohort	Stable COPD outpatients	Baseline	30 (27)	66.4 (10.3)	74.3 (66.5)	34.8 (13.6)	6 weeks
				12 month follow-up	30 (27)	66.4 (10.3)	74.3 (66.5)	34.8 (13.6)	6 weeks
Wilkinson et al., 2019 [11]	UK	Prospective cohort	Stable COPD outpatients	Year 1	127 (47)	66.8 (8.6)	47.0 {33.7–60.0}	46.4 (15.2)	Stable clinical condition
				Year 2	127 (44)	66.7 (8.7)	50.4 {34.0–60.0}	46.7 (14.6)	Stable clinical condition
Zalacain et al., 1999 [42]	Spain	Cross-sectional	Stable COPD outpatients	-	88 (0)	66.1 (7.2)	53.6 (14.9)	55.7 (12.9)	4 weeks
Zhang et al., 2010 [43]	China	Prospective cohort	Stable COPD outpatients	-	46 (17)	70.9 (5.6)	NA	51.8 (12.3)	6 weeks

Key: mean (SD); median [IQR]; mean {range}; mean, SEM *; Abbreviations: BAL, bronchoalveolar lavage; BLVR, bronchoscopic lung volume reduction; COPD, chronic obstructive pulmonary disease; EBV, endobronchial valve; FEV1, forced expiratory volume in 1 s; NA, data not available; PPM, potentially pathogenic micro-organism; PSB, protected specimen brushing.

**Table 3 biomedicines-10-00081-t003:** Sputum culture outcomes and prevalence of top five most commonly reported organisms.

Study (Author, Year)	Study Subgroup	Sampling Modality	No. of Patients Producing Sputum	No. of Sputum Samples Produced	Prevalence of PPM Positive Sputum, Percent (95% CI)	Prevalence of *H. influenzae* in Sputum, Percent (95% CI)	Prevalence of *M. catarrhalis* in Sputum, Percent (95% CI)	Prevalence of *S. pneumoniae* in Sputum, Percent (95% CI)	Prevalence of *P. aeruginosa* in Sputum, Percent (95% CI)	Prevalence of *S. aureus* in Sputum, Percent (95% CI)
Andelid et al., 2015 [12]	-	Spontaneous	40	40	50 (34–66)	13 (4–27)	3 (0–13)	5 (1–17)	3 (0–13)	0 (0–9)
Banerjee et al., 2004 [13]	-	Induced	67	67	40 (29–53)	21 (12–33)	15 (7–26)	13 (6–24)	2 (0–8)	2 (0–8)
Bogaert et al., 2004 [14]	-	Spontaneous	269	918	34 (31–38)	19 (17–22)	19 (17–22)	13 (11–15)	NA	NA
Cabello et al., 1997 [1]	PSB	PSB	18	18	28 (10–54)	11 (1–35)	0 (0–19)	11 (1–35)	0 (0–19)	6 (0–27)
	BAL	BAL	16	16	6 (0–30)	0 (0–20.6)	0 (0–21)	6 (0–30)	0 (0–21)	0 (0–21)
Einarsson et al., 2016 [15]	-	BAL	18	18	100 (82–100)	28 (10–54)	0 (0–19)	17 (4–41)	6 (0–27)	6 (0–27)
Fruchter et al., 2014 [16]	-	BAL	70	70	57 (45–69)	7.1 (2–16)	1 (0–8)	4 (1–12)	17 (9–28)	13 (6–23)
Garcia-Nunez et al., 2014 [17]	Moderate-to-severe disease	Spontaneous	8	8	63 (25–92)	13 (0–53)	25 (3–65)	13 (0–53)	38 (8–76)	NA
	Advanced disease	Spontaneous	9	9	78 (40–97)	33 (8–70)	0 (0–34)	11 (0–48)	22 (3–60)	NA
Hurst et al., 2005 [18]	Whole cohort	Spontaneous or induced	47	47	43 (28–58)	19 (9–33)	6 (1–18)	6 (1–18)	2 (0–11)	NA
Jacobs et al., 2018 [19]	-	Spontaneous	181	7464	28 (27–29)	14 (13–15)	6 (5–6)	6 (5–6)	8 (7–8)	NA
Jordan et al., 1976 [20]	BAL	BAL	19	27	52 (32–71)	22 (9–42)	0 (0–13)	NA	11 (2–29)	4 (0–19)
	Trans-tracheal aspiration	Trans-tracheal aspiration	11	15	33 (12–62)	20 (4–48)	0 (0–22)	NA	0 (0–22)	0 (0–22)
Khurana et al., 2014 [21]	Non-persistent sputum	Spontaneous or induced	13	13	8 (0–36)	8 (0–36)	0 (0–25)	0 (0–25)	0 (0–25)	0 (0–25)
	Persistent sputum	Spontaneous or induced	20	20	55 (32–77)	35 (15–59)	5 (0–25)	15 (3–38)	0 (0–17)	5 (0–25)
Marin et al., 2009 [22]	Baseline	Induced	40	79	73 (62–83)	35 (25–47)	5 (1–13)	0 (0–5)	NA	NA
	9 month follow-up	Induced	40	79	71 (60–81)	32 (22–43)	3 (0–9)	0 (0–5)	NA	NA
Marin et al., 2012 [23]	-	Spontaneous or induced	133	133	2 (22–38)	17 (11–24)	5 (2–10)	4 (1–9)	6 (3–12)	NA
Mika et al., 2018 [24]	-	BAL	20	20	30 (12–54)	15 (3–38)	10 (1–32)	10 (1–32)	NA	NA
Millares et al., 2014 [10]	Whole cohort	Spontaneous	14	14	86 (57–98)	29 (8–58)	14 (2–43)	14 (2–43)	36 (13–65)	0 (0–23)
Miravitlles et al., 2009 [25]	At randomisation	Induced	119	119	38 (29–47)	16 (10–24)	3 (1–8)	3 (1–7)	4 (1–10)	0 (0–3)
	Placebo 8 week follow-up	Induced	20	20	80 (56–94)	50 (27–73)	5 (0–25)	0 (0–17)	0 (0–17)	0 (0–17)
Miravitlles et al., 2010 [26]	-	Spontaneous or induced	119	119	49 (40–58)	18 (11–26)	3 (1–8)	3 (1–8)	4 (1–10)	1 (0–5)
Monso et al., 1995 [27]	-	PSB	40	40	33 (19–49)	15 (6–30)	3 (0–13)	8 (2–20)	3 (0–13)	3 (0–13)
Monso et al., 1999 [28]	-	PSB	41	41	22 (11–38)	12 (4–26)	NA	5 (1–17)	NA	NA
Patel et al., 2002 [4]	-	Induced	29	29	52 (33–71)	28 (13–47)	10 (2–27)	NA	10 (2–27)	NA
Riise et al., 1994 [29]	Without COPD	PSB	19	19	16 (3–40)	11 (1–33)	0 (0–18)	5 (0–26)	NA	0 (0–18)
	With COPD	PSB	18	18	11 (1–35)	0 (0–19)	0 (0–19)	11 (1–35)	NA	0 (0–19)
Seemungal et al., 2008 [30]	-	Spontaneous	69	69	52 (40–64)	32 (21–44)	4 (1–12)	9 (3–18)	NA	NA
Sethi et al., 2006 [31]	-	BAL	26	26	35 (17–56)	12 (3–30)	0 (0–13)	4 (0–20)	4 (0–20)	4 (0–20)
Sibila et al., 2014 [32]	-	PSB	37	37	27 (14–44)	14 (5–29)	5 (1–18)	5 (1–18)	0 (0–10)	0 (0–10)
Sibila et al., 2016 [33]	-	PSB	45	45	31 (18–47)	18 (8–32)	4 (1–15)	4 (1–15)	0 (0–8)	NA
Simpson et al., 2014 [34]	-	Induced	25	25	36 (18–58)	40 (0–20)	4 (0–20)	8 (1–26)	16 (5–36)	4 (0–20)
Simpson et al., 2016 [35]	-	Induced	59	59	24 (14–37)	5 (1–14)	12 (5–23)	NA	7 (2–17)	3 (0–12)
Singh et al., 2014 [36]	-	Spontaneous or induced	99	116	11 (9–22)	6 (3–12)	1 (0–5)	4 (1–10)	2 (0–6)	1 (0–5)
Sriram et al., 2018 [37]	-	BAL	27	27	37 (19–58)	22 (9–42)	NA	4 (0–19)	7 (1–24)	4 (0–19)
Trudzinski et al., 2018 [38]	-	BAL	64	64	47 (34–60)	9 (4–19)	2 (0–8)	6 (2–15)	5 (1–13)	6 (2–15)
Tumkaya et al., 2006 [39]	Exacerbations (<3/year)	BAL	20	20	55 (32–77)	0 (0–17)	0 (0–17)	10 (1–32)	NA	0 (0–17)
	Exacerbations (>3/year)	BAL	19	19	69 (44–78)	11 (1–33)	5 (0–26)	0 (0–18)	NA	5 (0–26)
Weinreich et al., 2007 [40]	-	BAL	53	53	43 (30–58)	23 (12–36)	4 (1–13)	25 (14–38)	4 (1–13)	4 (1–13)
Wilkinson et al., 2003 [41]	Baseline	Spontaneous or induced	30	30	53 (34–72)	30 (15–49)	10 (2–27)	10 (2–27)	10 (2–27)	NA
	12 month follow-up	Spontaneous or induced	30	30	57 (37–75)	23 (10–42)	23 (10–42)	0 (0–12)	10 (2–27)	NA
Wilkinson et al., 2019 [11]	Year 1	Spontaneous or induced	127	952	49 (46–52)	30 (27–33)	5 (4–7)	19 (16–21)	5 (4–7)	4 (3–6)
	Year 2	Spontaneous or induced	103	676	43 (39–47)	23 (19–26)	3 (2–5)	16 (13–19)	5 (3–7)	6 (5–9)
Zalacain et al., 1999 [42]	-	PSB	88	88	31 (21–41)	16 (9–25)	5 (1–11)	8 (3–16)	0 (0–4)	1 (0–6)
Zhang et al., 2010 [43]	-	Spontaneous	46	46	37 (23–53)	15 (6–29)	2 (0–12)	9 (2–21)	4 (1–15)	2 (0–12)

All prevalence data calculated as percent positive of total number of sputum samples. Data for the top five most commonly reported organisms shown. Abbreviations: BAL, bronchoalveolar lavage; COPD, chronic obstructive pulmonary disease; NA, data not available; PPM, potentially pathogenic micro-organism; PSB, protected specimen brushing.

## 4. Discussion

This systematic review and meta-analysis provides the first in-depth look at the colonisation of the airways of COPD patients in the stable state. Potentially pathogenic microbes were isolated from 41% of 11,862 sputum samples. Unsurprisingly, the most commonly reported organisms were *H. influenzae*, *M catarrhalis, S. pneumoniae, S. aureus,* and *P. aeruginosa.* Although *S. viridans* and *H. parainfluenzae* were not as commonly reported, they were equally as prevalent in sputum as the previous organisms, when authors sought them or reported data for them. The age, smoking history, and gender of the included patients were as expected for a general COPD population.

A large degree of heterogeneity was observed between studies (I^2^ = 94%) and was likely a result of variation in the reporting of bacteria between studies. Here, we report prevalence of author-defined PPMs, although definitions of a PPM may have differed between studies. For instance, Einarsson et al. (2016) deemed *Streptococcus viridans* as a PPM, and this was recovered in every sample within the study and hence gave a total prevalence of 100% [15]. While *S. viridans* is a common commensal of the upper respiratory tract, when infecting the lower airways it typically causes complicated parapneumonic effusions and/or empyema, and is not thought to cause acute exacerbation of COPD [44]. Interestingly, despite its recognition as an upper airway commensal, several studies here identified this organism with protected specimen brushing [1,27,28,42]. Along a similar line, it is possible that some pathogens may be under-represented through reporting practice. Several papers restricted their reporting to a predefined list of PPMs, and others may have restricted cultured organisms methodologically through selection of agar and growth conditions [11,14,19,25,26,30,43]. In this review we aimed to limit this effect through reporting only those PPMs that were mentioned specifically within the methods or results. In other words, we did not take the absence of evidence for evidence of absence. Alternatively, we could have pre-selected a defined list of PPMs for which we limited our data collection to, however this may have led to the under- or over-representation of certain bacteria and would have limited our investigation of the total prevalence in COPD patients. Some important organisms in airways disease, such as aspergillus, are not bacteria and hence were beyond the scope of this review. Future work should look to consider co-colonisation with bacterial and fungal pathogens and how this may affect the disease phenotype.

Another possible discrepancy amongst studies was in the definition of stability in COPD. Stability in COPD is poorly defined, and several definitions may exist across studies. In this review we chose to allow the authors’ definitions of COPD and have reported them here where available (Table 2). These definitions range from 15 days to 16 weeks, with several papers simply reporting patients as ‘clinically stable’ or ‘stable according to classical criteria’. This is an important topic as a longer stability period may imply certain characteristics within this population. Longer stability may mean a greater duration from the time of last antibiotic use, and may also imply a phenotype of a patient who is a less frequent exacerbator and thus more likely to be enrolled. It is likely that these differing populations also contribute to the large degree of heterogeneity across studies.

Finally, we must discuss the definition of colonisation. The majority of studies included here reported only single sputum cultures performed during a single period of stability. However, results from the Acute Exacerbation and Respiratory Infections in COPD (AERIS) cohort, who had sputum samples collected monthly, show that colonisation is dynamic and year-to-year variation in colonisation may affect exacerbation rate [11]. Single sputum samples, especially spontaneous sputum, are vulnerable to variations over time, varying lung regions, and varying sample quality. Colonisation also does not happen in isolation and is instead a dynamic occurrence. Thereby, to truly reflect and understand “colonisation” of the airways, repeat samples over a defined period should be taken, which has not often been done to date, emphasising the importance of this for future work.

The prevalence of PPM-positive sputum defined here includes both single and multiple organisms within a binary definition of prevalence. It is likely that not only the presence of PPMs in the sputum of stable COPD patients affects the clinical phenotype, but also the composition and number of these PPMs. Exacerbations of COPD may be triggered by the introduction of a new bacterial strain or species, by the sudden increase in (colonising) bacterial load or by the mutation of a pre-existing bacterial strain [45]. For instance, bacteria may undergo antigenic changes which allow it to evade the host immune response, replicate and cause inflammation [46]. In the paper by Jacobs et al. (2018), co-colonisation with two or more of *H. influenzae*, *M. catarrhalis*, *S. pneumoniae*, and *P. aeruginosa* was identified in 5% of sputum samples from stable COPD patients [19]. A complex inter-relationship was found to exist between these organisms with the presence of some positively or negatively associated with the presence of others. Importantly, antimicrobial therapy in these patients was associated with suppression of all species except *P. aeruginosa*, an organism commonly associated with a clinically worse COPD phenotype. Targeted therapies not considering the wider picture of co-colonisation may have inadvertently deleterious effects.

We found a significant, although weak, association between higher prevalence of PPMs and FEV1 (used as a proxy marker of disease severity). The contribution of colonisation to disease progression is not yet clear but it is likely to be an important component of the “vicious cycle hypothesis”, whereby worsening of disease further increases susceptibility to colonisation [47].

What seems likely is that bronchial colonisation in the stable state contributes to the overall phenotype of COPD patients, and what may not classically be considered a PPM may still be relevant to the clinical picture [4,48,49]. Future studies may wish to consider more broadly capturing the respiratory microbiome in COPD, rather than limiting to those considered PPMs. Further, microbiome studies detect non-viable microorganisms that may also trigger the immune system and inflammatory response within the airways.

Our systematic review was limited to studies that performed culture as a way of detecting colonisation. Inclusion of studies that performed microbiome analysis would have captured an entire “picture” of what is present in the lower airways of stable COPD patients (including non-viable and atypical bacteria, undetectable via culture) but would have added a large element of complexity and heterogeneity to our results.

When interpreting this study, it is important to consider that mean prevalence is reported as a percentage of samples in individual studies, rather than a percentage of patients. Whilst this still gives a general representation of prevalence in a COPD population, some patients may have given repeat samples and hence biased the results.

Across the included studies, we found a total of 23 separate microbes reported as PPMs and found in the stable state. With such a varied and prevalent collection of colonising bacteria it may be that future work will identify those that contribute more significantly to the exacerbation phenotype, and these could be targeted for personalised therapy. Given the issue of antimicrobial resistance, as well as the economic burden, it is critical that antibiotics are only prescribed where they will provide a clinical benefit. There is difficulty, however, in determining the exact role and clinical impact of colonisation in COPD. Sputum samples are not routinely collected in stable COPD patients and are typically only of clinical interest during the onset or course of an exacerbation. Moreover, not all COPD patients have chronic bronchitis and therefore sampling in the stable state may be restricted. A potential way of enhancing clinical decisions on antibiotic usage may be with the use of the sputum colour chart where purulent (green) sputum has shown 94.4% sensitivity for the yield of bacterial load [50]. Patients who do produce sputum in the stable state could be encouraged to provide samples during colour change, but in the absence of symptoms of exacerbation. Targeted therapy could then be determined based on the patient’s stable microbiota ahead of exacerbation, rather than as a reactionary measure during acute illness. This may avoid the need for these patients to be started on long-term antimicrobial prophylaxis, reducing the potential for resistance and avoiding unwanted side effects.

## 5. Conclusions

Bacterial colonisation has been previously associated with increased inflammation, worsening of symptoms and increased frequency of exacerbations. Here, we found that 44% of samples collected from stable COPD patients are positive for a PPM, with *Haemophilus influenzae* being the most commonly isolated organism. However, data on many important clinical outcomes in relation to colonisation was poorly reported. Hence, there exists a cohort of COPD patients who are colonised with PPMs, but the clinical significance of this is yet to be elucidated.

## Figures and Tables

**Figure 1 biomedicines-10-00081-f001:**
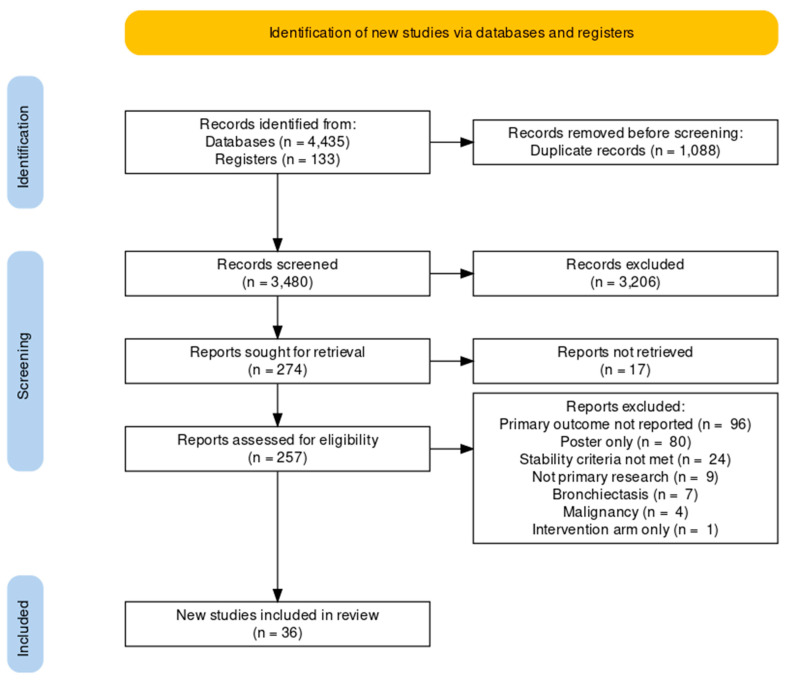
Preferred Reporting Items for Systematic Reviews and Meta-Analyses (PRISMA) flow chart of study selection and inclusion.

**Figure 2 biomedicines-10-00081-f002:**
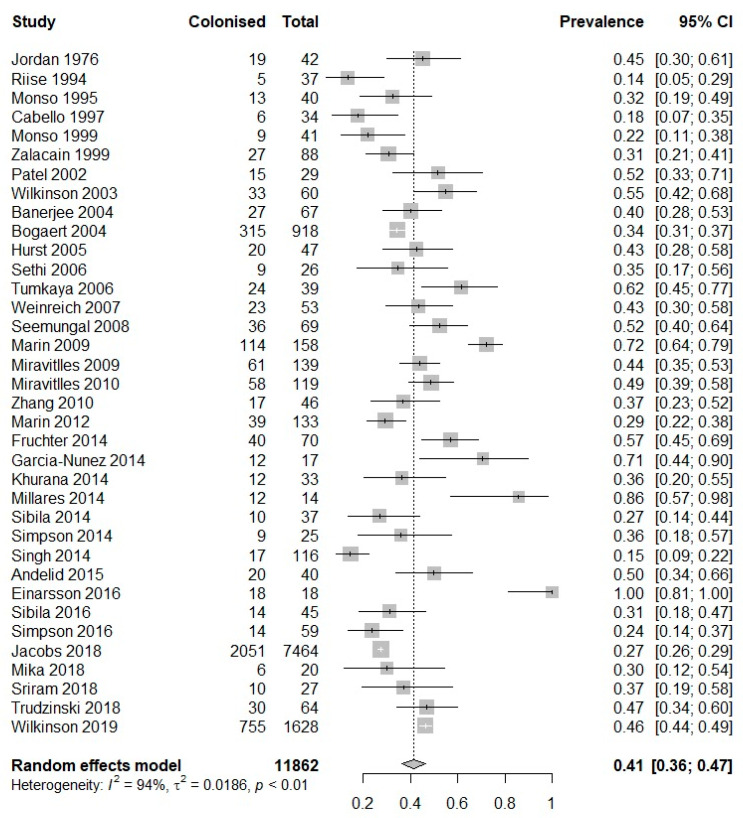
Meta-analysis of prevalence of potentially pathogenic microorganisms in quantitative cultures from stable COPD patients.

**Figure 3 biomedicines-10-00081-f003:**
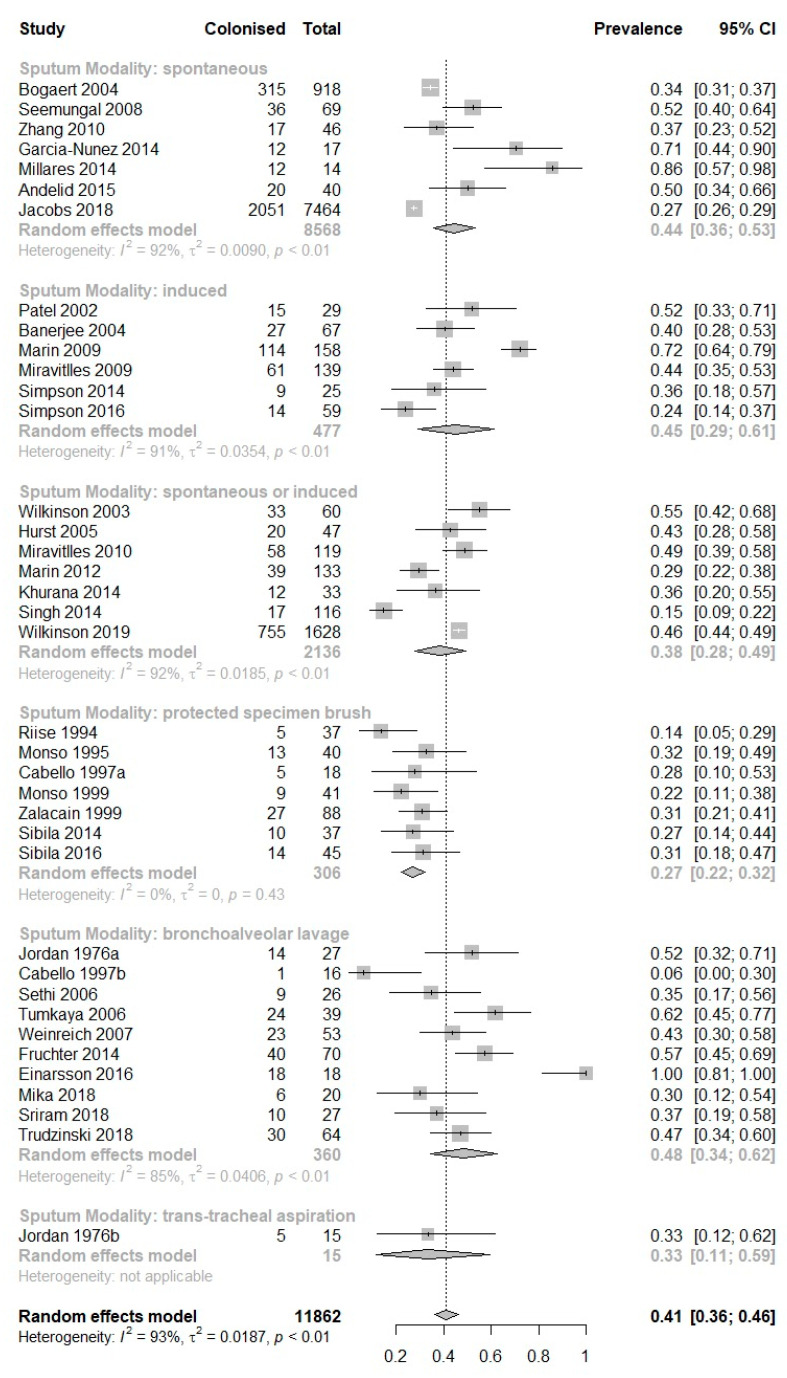
Meta-analysis of prevalence of potentially pathogenic microorganisms in quantitative culture-sub-grouped by quantitative culture modality.

**Figure 4 biomedicines-10-00081-f004:**
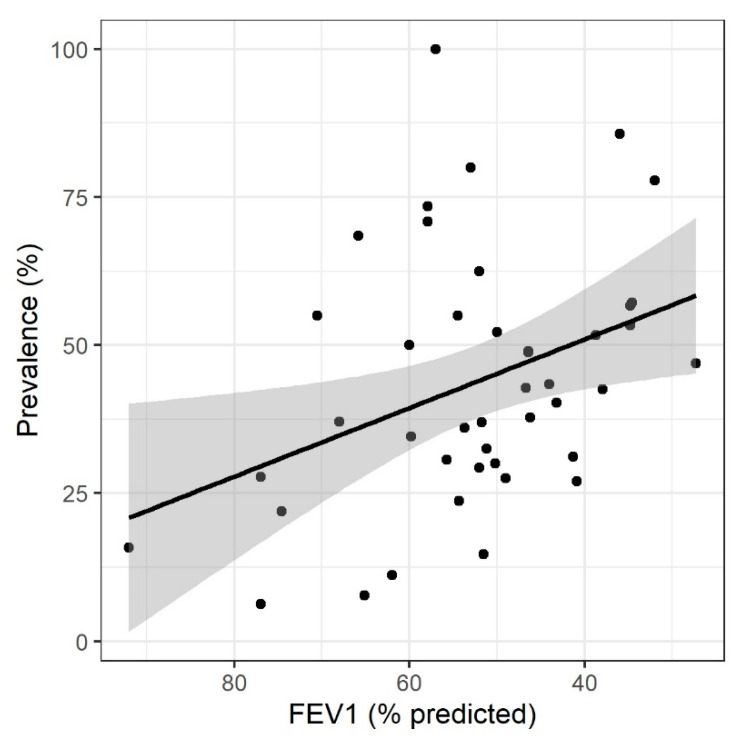
Scatterplot of bacterial colonization prevalence by declining mean or median FEV1 (% predicted). Each individual data point represents one study. Two studies (Bogaert 2004 and Jordan 1976) did not report FEV1 data, and as such are excluded here. R^2^ = 0.136, *p* = 0.015. FEV1, forced expiratory volume in 1 s.

**Figure 5 biomedicines-10-00081-f005:**
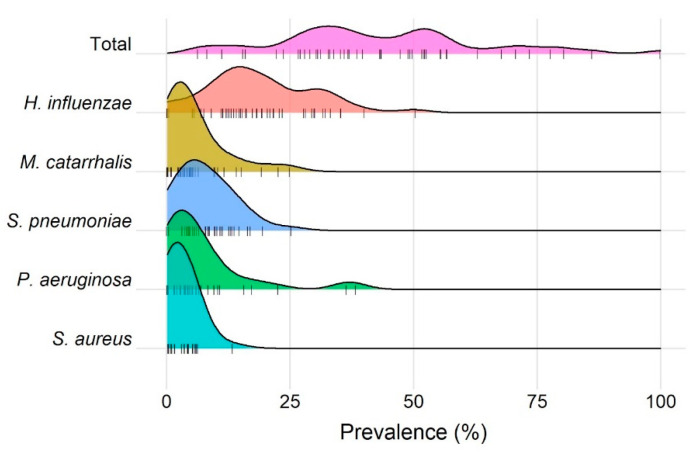
Ridgeline plot of the distribution of studies’ mean prevalence of potentially pathogenic microorganisms. Distributions of total prevalence, and prevalence of the top five most commonly reported organisms, are shown with each study contributing a single datapoint. Number of studies reporting each organism were: *H. influenzae* (*n* = 36), *M catarrhalis* (*n* = 34)*, S. pneumoniae* (*n* = 33)*, P. aeruginosa* (*n* = 27), and *S. aureus* (*n* = 24).

**Table 1 biomedicines-10-00081-t001:** Summary of outcomes and data collection points. CAT, COPD assessment test; COPD, chronic obstructive pulmonary disease; FEV1, forced expiratory volume in 1 s; GOLD, Global Initiative for Chronic Obstructive Lung Disease; mMRC, modified medical research council; PPM, potentially pathogenic microorganism; PSB, protected specimen brushing; SGRQ, St. George’s Respiratory Questionnaire.

Outcomes		Data Collection Points
Demographics		Age; sex; alpha-1 antitrypsin status; stability period; smoking status and pack years
Primary Outcome	Determine prevalence of bacterial colonisation in stable-state COPD	Number of patients that produced a sample; number of samples collected; number of positive cultures; individual bacteriology (number positive for individual PPMs)
Secondary Outcomes	Assess the relationship between sampling modality and colonisation	Sampling modality (spontaneous, induced, PSB, bronchoscopy, trans-tracheal aspiration)
	Assess relationship between bacterial colonisation and disease phenotype	FEV1; FEV1 category by GOLD criteria; quality of life (SGRQ/CAT/mMRC); exacerbation frequency; hospitalisation rate; mortality rate

## Data Availability

The data presented in this study are available on request from the corresponding author.

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
