# Peer review of "A Systematic Review and Meta-Analysis of the Prevalence and Impact of Pulmonary Bacterial Colonisation in Stable State Chronic Obstructive Pulmonary Disease (COPD)"

_biomedicines, 2021, doi:10.3390/biomedicines10010081_

Round 1
Reviewer 1 Report
In introduction, in the part about prophylactic treatment , the references are missing, and some data, especially about use of azitromicin, are not completelly correct.
Author Response
Dear reviewer,
Many thanks for your comments. We have altered the citations in the introduction paragraph regarding the use of prophylactic antibiotics in COPD to better reflect the origin of our points. Citation 7 was for a large Cochrane review and meta-analysis from which we derived our points regarding variability of drug choice, efficacy of treatment, and recommendation for considering the role of colonisation. We have now included this citation for both these sentences.
We are unsure exactly what is incorrect regarding the use of azithromycin for antimicrobial prophylaxis in COPD, however we are happy to correct or address this as necessary if you are able to be more specific.
Kind regards,
Michael
Reviewer 2 Report
The authors stress the usefulness of sputum microbiology in acute COPD exacerbations for targeted prevention, and performed a systematic review and meta-analysis. The most commonly reported organisms were H. influenzae, M catarrhalis, S. pneumoniae, S. aureus, and P. aeru-ginosa. The decline in lung function was weakly correlated with the prevalence of potentially pathogenic microorganisms. They concluded that the airways of COPD patients are colonized by potentially pathogenic microorganisms during the stable state in nearly half of the patients and that targeted microbial therapy to prevent COPD exacerbations should carefully consider the stable and exacerbated microbiome.
The review is very interesting and I have only a question for you: Have you found data on microbiota-related chronic bronchopulmonary aspergillosis? Especially in categories of patients with other underlying diseases?
Mult Scler. 2020 Jan;26(1):123-126. doi: 10.1177/1352458518813110. Epub 2019 Mar 18. PMID: 30882274.
Author Response
Dear reviewer,
Many thanks for your comments. As this paper was focused on bacterial colonisation it was deemed that fungal colonisation was beyond the scope of this review. We agree that aspergillus is an important organism in COPD and would be interested to see future work which considers fungal and bacterial co-colonisation. As mentioned, this would be particularly interesting to consider alongside data regarding patient comorbidities. We have added to the relevant section of the discussion with a recommendation for this work to be considered in the future, however we do not think it would be appropriate to address this here given the focus of this review.
Many thanks,
Michael